# Simultaneous loss of interlayer coherence and long-range magnetism in quasi-two-dimensional PdCrO$_2$

S. Ghannadzadeh[1,2,3], S. Licciardello[1,2], S. Arsenijević[1,2,4], P. Robinson[1,2], H. Takatsu[5,6], M.I. Katsnelson[2] & N.E. Hussey[1,2]

In many layered metals, coherent propagation of electronic excitations is often confined to the highly conducting planes. While strong electron correlations and/or proximity to an ordered phase are believed to be the drivers of this electron confinement, it is still not known what triggers the loss of interlayer coherence in a number of layered systems with strong magnetic fluctuations, such as cuprates. Here, we show that a definitive signature of interlayer coherence in the metallic-layered triangular antiferromagnet PdCrO$_2$ vanishes at the Néel transition temperature. Comparison with the relevant energy scales and with the isostructural non-magnetic PdCoO$_2$ reveals that the interlayer incoherence is driven by the growth of short-range magnetic fluctuations. This establishes a connection between long-range order and interlayer coherence in PdCrO$_2$ and suggests that in many other low-dimensional conductors, incoherent interlayer transport also arises from the strong interaction between the (tunnelling) electrons and fluctuations of some underlying order.

[1] High Field Magnet Laboratory (HFML-EMFL), Faculty of Science, Radboud University, Toernooiveld 7, 6525ED Nijmegen, The Netherlands. [2] Institute of Molecules and Materials, Faculty of Science, Radboud University, 6525 AJ Nijmegen, The Netherlands. [3] Oxford Instruments NanoScience, Tubney Woods, Abingdon, Oxfordshire OX13 5QX, UK. [4] Dresden High Magnetic Field Laboratory (HLD-EMFL), Helmholtz-Zentrum Dresden-Rossendorf, 01328 Dresden, Germany. [5] Department of Physics, Tokyo Metropolitan University, Tokyo 192-0397, Japan. [6] Department of Energy and Hydrocarbon Chemistry, Graduate School of Engineering, Kyoto University, Kyoto 615-8510, Japan. Correspondence and requests for materials should be addressed to S.G. (email: s.ghannadzadeh@physics.oxon.org) or to N.E.H. (email: n.e.hussey@science.ru.nl).

Many correlated metals, such as cuprates, ruthenates and iron-based superconductors have highly anisotropic electronic properties, often resulting in an interlayer conductivity that is incoherent, even at low temperatures. Despite intense theoretical and experimental investigation, the origin of this incoherence is unknown, though it is likely due to some combination of strong electron correlations, the lamellar crystalline structure and/or proximity to an ordered phase. In quasi-two-dimensional (Q2D) systems that are both metallic and magnetic, the (weak) electronic coupling between the conducting planes can also influence the interlayer exchange coupling $J'$. Although the effect of electronic coupling on the magnetic ordering—for example via the Ruderman–Kittel–Kasuya–Yosida (RKKY) interaction[1–4]—has been widely studied[5,6], there has been very little experimental exploration of the effect of magnetism itself on the interlayer electronic coupling and on the electronic dimensionality.

PdCrO$_2$ is a rare example of a highly metallic 2D triangular antiferromagnet. As shown in Fig. 1a, PdCrO$_2$ has a delafossite structure, consisting of stacked layers of highly conducting Pd layers sandwiched between planes of Mott-insulating CrO$_2$ (refs 7,8). The Cr ions have a localized spin of 3/2 and are highly frustrated, forming a non-coplanar and non-collinear antiferromagnetic 120° helical structure at $T_N = 37.5$ K (refs 7,9–12). Quantum oscillation[8,13] and angle-resolved photoemission studies[14] have mapped out the Fermi surface (FS) of PdCrO$_2$. At $T > T_N$, the FS consists of a single six-fold symmetric electron pocket centered at $\Gamma$ and is mostly derived from the $4d^9$ electrons[15]. At $T_N$, band-folding due to the $Cr^{3+}$ spin ordering leads to formation of a $\sqrt{3} \times \sqrt{3}$ supercell and results in FS reconstruction into the more complex FS, indicating a strong coupling between the magnetism and the conduction electrons. A recent magnetothermopower study also revealed a strong interaction between the $4d^9$ electrons and the short-range spin correlations persisting well above $T_N$ (ref. 10).

Here we report the observation that the interlayer coherence of the Pd electrons is lost upon transition from the magnetically ordered to the paramagnetic regime above $T_N$. Through high-field angle-dependent magnetoresistance (ADMR) measurements at temperatures above and below $T_N$, we chart the evolution of the so-called Hanasaki coherence peak—one of the most definitive and sensitive probes of electronic coherence[16–21]—as a function of temperature. We show that the coherence peak is fully suppressed just above $T_N$, implying a close correlation between the magnetic order on the Cr sites and the coherence of the Pd electronic states. This finding raises the question of whether it is the interlayer electron coherence that renormalizes $J'$ and hence $T_N$, or whether it is the melting of the magnetism that induces a dimensional crossover in the conduction electrons. We extract the interlayer hopping parameter $t_\perp$ from the Hanasaki peak, and through comparison with the isostructural non-magnetic PdCoO$_2$, we argue that it is the loss of long-range magnetic order that ultimately decouples electronically the conducting planes.

## Results

**Angle-dependent magnetoresistance.** Figure 2b shows the $c$-axis magnetoresistance $\rho_c(\theta)$ of PdCrO$_2$ at $T = 4.2$ K in magnetic fields of 15 and 30 T as the sample is rotated around the polar axis, from the field perpendicular ($\theta = 0°$, $\mathbf{H}\|[001]$) to parallel ($\theta = 90°$, $\mathbf{H}\|[110]$) to the conduction planes. The data show a broad near-sinusoidal background with a minimum at $\theta = 0°$ (the Lorentz force-free configuration), on which is superimposed a series of complex ADMR oscillations (AMROs). These peaks, also known as Yamaji oscillations, occur at certain orientations of the magnetic field whenever the interplane electron velocity, when averaged over its corresponding cyclotron orbit[22], is minimized[23]. The angular location of these peaks is governed by the relation $dk_F^\| \tan\theta = \pi(n - 1/4)$ for each FS pocket, where $n$ is an integer, $d = 6.03$ Å is the interplanar distance and $k_F^\|$ is the projection of the Fermi wave-number on the conducting plane[23]. While the multi-component nature of the FS in PdCrO$_2$ leads to some ambiguity when assigning individual peaks to a specific pocket, some peaks can still be identified. As an example, the series of peaks indicated by arrows in Fig. 2b correspond to a pocket of radius $k_F^\| = 0.57(3)^{-1}$ (see inset of Fig. 2b), consistent with that of the $\gamma$ pocket identified by Shubnikov–de Haas oscillations[8,13]. The presence of polar AMRO, however, is not by itself evidence for a fully coherent three-dimensional (3D) FS (ref. 17).

**Temperature dependence of the interlayer coherence peak.** We turn now to the most dominant feature of the ADMR data, namely the sharp peak observed when the field is applied exactly parallel to the conducting planes ($\theta = 90$). This peak is unambiguously resolved at fields as low as 10 T (see Supplementary Fig. 2b), with the variation in field having no effect on the width

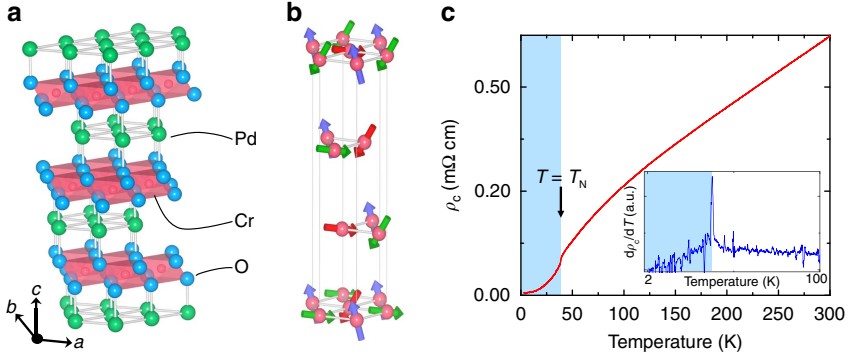

**Figure 1 | Magnetic ordering in PdCrO$_2$.** (**a**) Crystal structure of PdCrO$_2$, with lattice parameters $a = b = 2.930$ Å and $c = 18.087$ Å (ref. 7). The green, blue and red spheres represent the Pd, O and Cr atoms, respectively. The red-shaded planes represent the sides of the edge-shared CrO$_6$ octahedra. (**b**) One of the possible solutions for magnetic structure of the antiferromagnetically ordered phase below $T_N = 37.5$ K, showing a non-coplanar spin structure[9]. The arrows represent the Cr spins, with arrows of the same color representing spins in the same spin sublattice group (see ref. 9 for details). For clarity only the Cr atoms are shown. (**c**) The temperature dependence of the interlayer $c$-axis resistivity $\rho_c$, which shows a sub-linear temperature dependence in the paramagnetic regime with a sharp cusp at $T_N$, resulting in a $\rho_c(T)$ that rises faster than $T^2$ in the FS reconstructed phase. Inset: The derivative $d\rho_c/dT$ of the same resistivity curve, highlighting the sharp peak at the transition temperature. The shaded and unshaded regions indicate the coherent and incoherent regimes, respectively. Crystallographic drawings produced using VESTA[37].

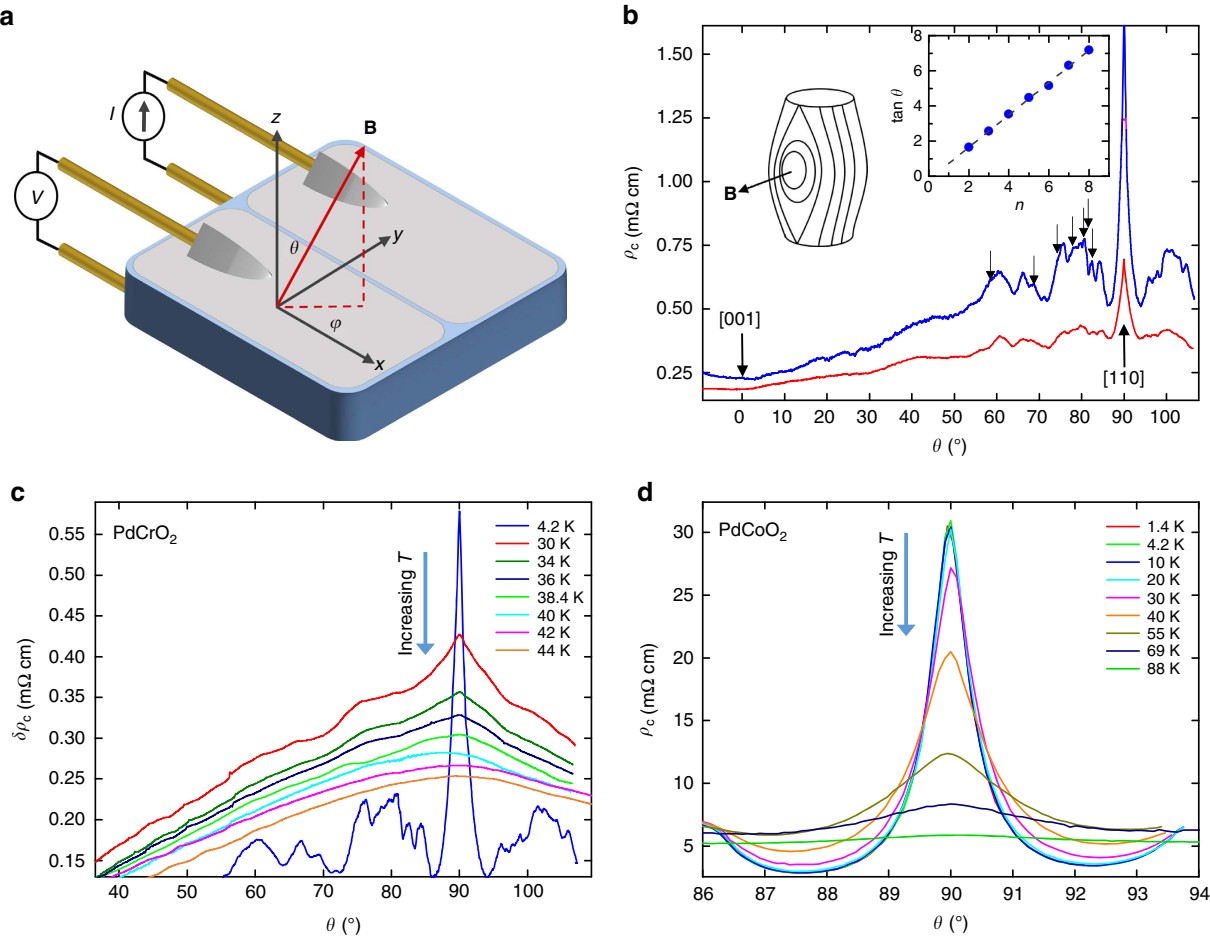

**Figure 2 | ADMR at high fields.** (**a**) Schematic diagram of the four-contact setup used to measure $c$-axis resistivity, showing the definition of the polar ($\theta$) and azimuthal ($\phi$) angles. (**b**) Evolution of $\rho_c$ during a polar rotation at 15 T (red line) and 30 T (blue line), at 4.2 K. The direction of the applied field at 0° and 90° are given. The top right inset shows $\tan\theta$ for the peaks indicated by the arrows, forming a straight line as expected for Yamaji oscillations. Fits (dashed line in the inset) to this give $k_f = 0.57(3)$ Å$^{-1}$, consistent with that found for the $\gamma$ orbits from quantum oscillation measurements[13]. The left inset is a schematic showing the coherent FS orbits that are formed when the field is applied parallel to the crystal planes. (**c,d**) The reduction in the amplitude of the $c$-axis coherence peak at $\theta = 90°$ as a function of temperature for PdCrO$_2$ ($\mu_0 H = 30$ T) and for its non-magnetic isostructural analogue PdCoO$_2$ ($\mu_0 H = 35$ T)[28]. For clarity, the PdCrO$_2$ data is plotted as $\delta\rho_c = \rho_c(\theta) - \rho_c(\theta = 0)$. The 4.2 K data for PdCrO$_2$ have been scaled by 0.43. The PdCoO$_2$ data are reproduced with kind permission from Kikugawa et al.[28]

of the peak, only its amplitude. First discussed in depth by Hanasaki et al.[16], the peak arises due to formation of closed orbits parallel to the conducting planes in a warped FS column, as shown schematically in the inset of Fig. 2b (Note that although open orbits can also contribute to the interlayer conductivity for in-plane fields, they do not lead to a peak in the resistance[19]). Thus, in contrast to polar AMRO, the Hanasaki peak is a direct signature of interlayer coherence and implies the existence of a FS that extends in all three dimensions[16–21,24].

Figure 2c shows the evolution of the Hanasaki peak as the temperature is raised through the magnetic transition. (It should be stressed that the magnetic field has a negligible effect on the value of $T_N$, at least at the field strengths employed in this study[8]). With increasing temperature, the amplitude of the coherence peak gradually diminishes, until eventually, only the broad sinusoidal background is visible. In order to follow its evolution more closely, we plot in Fig. 3a the temperature dependence of $d\rho_K(\theta)/d\theta$, where $\rho_K(\theta)$ is the Kohler-scaled resistivity $\rho_c(\theta)/\rho_{c,0}$ with $\rho_{c,0}$ being the zero-field $c$-axis resistivity for that particular temperature. In this plot, a change in gradient from positive to negative at 90° indicates the presence of the coherence peak (as explained in the Supplementary Note 2, a near-sinusoidal

background has been subtracted first from the raw data before differentiation). A change in gradient is indeed seen for all temperatures below $T_N$, but not at 40, 42 or 44 K. The sharpness of the peak can be quantified by looking at the magnitude of the second-derivative of $\rho_K(\theta)$. This quantity, plotted in Fig. 3b, is found to decrease almost linearly with increasing temperature, reaching zero just above $T_N = 37.5$ K. In other words, the coherence peak is found to be fully suppressed above the magnetic ordering temperature, implying that the $c$-axis FS warping becomes ill-defined and the original 3D FS is transformed into a stack of 2D sheets (see insets of Fig. 3b).

## Discussion

The near-coincidence of the loss of the Hanasaki peak and $T_N$ raises the intriguing conundrum of whether it is the change in electronic coherence which determines the value of $T_N$ in PdCrO$_2$, or conversely, if it is the loss of long-range order at $T_N$ that causes the interlayer hopping to become incoherent. In quasi-1D PrBa$_2$Cu$_4$O$_8$, a field-induced reduction in the dimensionality of the chain carriers drives a spin–flop transition of the local moments on the Pr sites[25]. There, the persistence of the

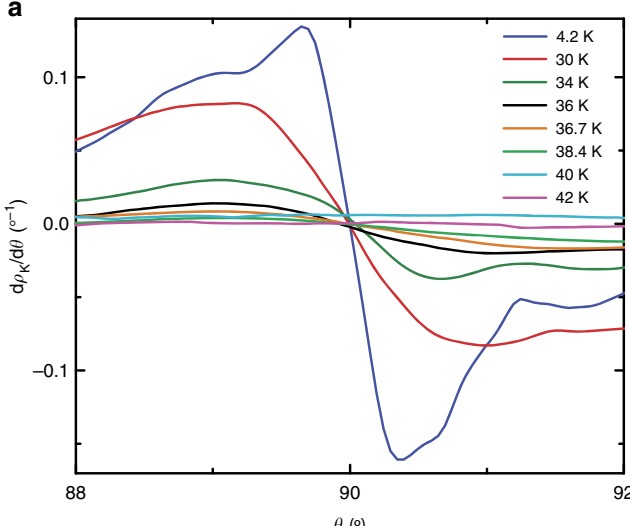

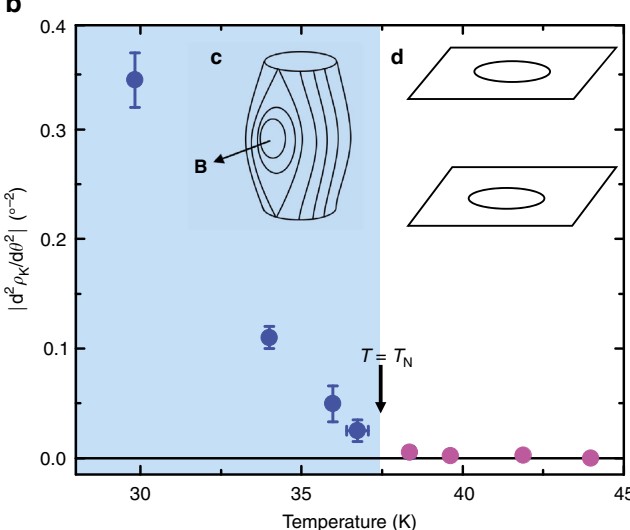

**Figure 3 | Loss of electronic coherence at $T_N$.** (**a**) Evolution of $d\rho_K/d\theta$ as a function of the polar angle for a range of temperatures above and below $T_N$. The 4.2 K curve is scaled by 0.03. (**b**) $|d^2\rho_K/d\theta^2|$ at $\theta = 90°$, showing the reduction in the sharpness of the coherence peak as temperature is increased. The shaded and unshaded backgrounds indicate the coherent and incoherent regimes, respectively. The temperature error bar is given by the temperature drift during the rotation, while the $|d^2\rho_K/d\theta^2|$ error bar is given by the maximum uncertainty introduced in the peak amplitude due to the necessary smoothing of the $d\rho_K/d\theta$ curve. The error bars are not shown if they are smaller than the data points. In both figures the broad sinusoidal background was subtracted prior to differentiation, as described in the Supplementary Note 2. Inset: representation of a 3D FS in the coherent regime (**c**), and a Fermi surface that is only defined in two-dimensions in the incoherent regime (**d**).

dimensional crossover beyond $T_N$ (ref. 26) confirms its key role in the spin–flop transition, the latter being attributed to a change in the effective RKKY interaction $J_{RKKY}$ induced by the corresponding reduction in $t_\perp$ of the mediating chain carriers. In PdCrO$_2$, a similar renormalization of $J_{RKKY}$ (via $t_\perp$) could also act to destabilize the long-range order and thereby renormalize $T_N$ to a value far below the Curie–Weiss temperature $\Theta_W$.

To determine which is the dominant effect in PdCrO$_2$, we need to compare first the energy scales that define the interlayer

coherence. Interlayer conductivity is assumed to become incoherent once the interlayer hopping integral $t_\perp$ becomes less than other relevant energy scales, such as temperature $k_B T$ (in which case, the $c$-axis warping becomes ill-defined) or the intraplanar scattering rate $\hbar/\tau$ (in which case, the individual carriers are scattered many times within the plane before tunnelling to an adjacent plane)[17,27]. The interlayer transfer integral can be obtained from the ADMR via the relation

$$t_\perp = \frac{\hbar^2 k_F}{4dm^*}\Delta\theta, \qquad (1)$$

where $m^*$ is the effective mass and $\Delta\theta$ is the full width of the coherence peak as measured from its base[16]. Before comparing the various energy scales in PdCrO$_2$, we first consider the case of isostructural PdCoO$_2$, which has an identical crystal structure and Fermiology to PdCrO$_2$ (above $T_N$) but has a non-magnetic ground state. The temperature evolution of the Hanasaki coherence peak in PdCoO$_2$ is shown in Fig. 2d (ref. 28). From its width, we obtain $t_\perp \approx 17$ meV $= 200$ K, in good agreement with the value ($= 21$ meV) obtained from a recent quantum oscillation study[15]. The coherence peak in PdCoO$_2$ persists to temperatures of order 90 K, consistent with these estimates for $t_\perp$. For PdCrO$_2$, we find $\Delta\theta = 8.0(5)°$, and assuming that the coherence peak is dominated by the largest non-breakdown orbital ($\gamma$) for which $k_F = 0.57(3)$ Å$^{-1}$ and $m^* = 1.37(2)m_e$ (ref. 13), we obtain $t_\perp \approx 18(1)$ meV $= 210(20)$ K, that is, a very similar magnitude to that found in PdCoO$_2$, as reflected in their comparable resistive anisotropies. Thus, despite the similarity in the $t_\perp$ magnitudes, the coherence peak in PdCrO$_2$ vanishes at a significantly lower temperature.

We can also estimate $\hbar/\tau$ just below $T_N$ from the magnitude of the in-plane resistivity and find $\hbar/\tau(T_N) \approx 0.8(1)$ meV (see Supplementary Note 3). Correspondingly, $\omega_c\tau \approx 3$ at 30 T and 37.5 K. In the quasi-2D organic superconductor $\kappa$-(ET)$_2$Cu(NCS)$_2$, the coherence peak itself was found to survive down to $\omega_c\tau$ values of order 1 (ref. 24), while in the high-$T_c$ cuprate Tl$_2$Ba$_2$CuO$_{6+\delta}$, polar AMRO have been observed down to $\omega_c\tau \sim 0.15$ (ref. 29). More concretely, it is instructive to compare directly the absolute magnitude of the resistivity in PdCrO$_2$ and in PdCoO$_2$ at the corresponding temperatures where the Hanasaki peak is found to vanish. Given that the carrier densities (and their effective masses $\sim 1.5m_e$) are essentially identical above 37.5 K (refs 13,15), the ratio of their resistivities should correspond to the ratio of their scattering rates. According to Hicks et al.[13], the resistivity in PdCrO$_2$ at $T_N$ is a factor of two larger than that of PdCoO$_2$ at 90 K, implying that the $\omega_c\tau$ value in PdCrO$_2$ when the Hanasaki peak vanishes is only half the corresponding value in PdCoO$_2$. Consequently, the suppression of the Hanasaki peak does not appear to be correlated with the carrier lifetime reaching a certain threshold. Moreover, given that both $k_B T$ and $\hbar/\tau$ are almost one order of magnitude smaller than $t_\perp$ at $T = T_N$, there is no obvious reason a priori why the coherence peak in PdCrO$_2$ should vanish beyond $T_N$. Therefore we conclude that it is not the change in electronic coherence which determines the value of $T_N$ in PdCrO$_2$, but that conversely, it is the loss of long-range magnetic order that induces the dimensional crossover of the conduction electrons and causes the interlayer hopping to become incoherent.

PdCrO$_2$ is an anisotropic-layered antiferromagnet with an interlayer exchange interaction $J'$ that is much smaller than the in-plane interaction $J$. This leads to the existence of a broad temperature range above $T_N$, $T_N < T < \Theta_W \approx 500$ K within which short-range antiferromagnetic fluctuations persist[7,9,11]. Such a state can be described via self-consistent spin-wave theory[30]. Analysis of the corresponding equations (see equations (20–23) of ref. 30) shows that whereas the in-plane correlation length

remains much larger than interatomic distance up to $T = \Theta_W$, the interplane correlation length $\xi_c$ becomes comparable to the $c$-axis lattice spacing $d$ at a much reduced temperature within $T_N/\Theta_W$ of the Néel ordering temperature,

$$\frac{T}{T_N} - 1 \approx \frac{T_N}{\Theta_W} \approx \frac{1}{\ln J/J'}. \qquad (2)$$

Once $\xi_c < d$, the magnetic coupling becomes strongly fluctuating. For $PdCrO_2$ the estimate of the right-hand side is $< 0.1$. A relatively compact explicit expression can be found in the limit of classical spins (which only effects numerical factors of the order of one):

$$\frac{T}{T_N} - 1 = \frac{1}{\ln J/J'} \ln\left(1 + \kappa\sqrt{1 + \frac{\kappa^2}{4}} + \frac{\kappa^2}{2}\right), \qquad (3)$$

where $\kappa = d/\xi_c$. Note that this expression is meaningful only for $\kappa < 1$, that is, very close to $T_N$. The motion of electrons in an in-plane magnetic field becomes incoherent at an even smaller $\kappa \sim d/r_c \ll 1$ where $r_c$ is the cyclotron radius along the $c$ axis. This implies that enhanced scattering of electrons off the spin fluctuations makes the interlayer electron motion incoherent even very close to the Néel temperature, that is, it is the magnetism that suppresses the electronic coherence at $T \approx T_N$ and not vice versa. Importantly, short-range magnetic order within the plane survives until $T \sim \Theta_W$.

Our analysis follows from the Heisenberg model where exchange interactions are considered as fixed parameters. By including an RKKY-type interaction in the calculation, one might expect that the incoherence of electron motion along the $c$-axis should in turn lead to a reduction in the effective RKKY coupling and a decrease in $J'$, thereby amplifying the effect discussed above. However, the RKKY interaction itself does not appear to be the driving force for setting $T_N$.

It will be interesting to explore whether a similar relationship between interlayer coherence and long-range order exists in other metallic antiferromagnets, such as $AgNiO_2$ or $Na_xCoO_2$, where a highly anisotropic electronic state co-exists with frustrated local moment magnetism[31,32]. More generally, the present finding may also have important implications for our understanding of interlayer decoherence in a host of other low-dimensional systems such as underdoped cuprates, ruthenates, iridates or Fe-based superconductors where short-range spin and/or charge fluctuations proliferate over a wide range of their respective phase diagrams. Looking further ahead, it also raises the prospect of bespoke electronic dimensionally control via tuning of the magnetism, for example through a combination of conduction metal layers and coordination polymer magnets, whose $J$ and $J'$ are highly tunable[33,34].

## Methods

**Crystal synthesis and selection.** Single crystals of $PdCrO_2$ were grown using a flux method, as described in ref. 7. A number of samples were contacted for standard four-contact transport measurements along the $c$ axis. To ensure optimal quality of the electrical contacts, DuPont 6838 conductive silver paste was used to contact the gold wires to the sample. The contacts were then cured in an Oxygen atmosphere. The evolution of the $c$-axis resistance $\rho_c(T)$ upon cooling from 300 to 2 K was measured using a Cryogen Free Measurement System, see Supplementary Fig. 1. The highest quality sample, with a residual resistivity ratio $\rho_c(300\,K)/\rho_c(2\,K)$ of 108 and dimensions of $\sim 0.6 \times 0.6 \times 0.2\,mm^3$, was chosen for the ADMR measurements.

**Angle-dependent magnetoresistance measurements.** All measurements were carried out at the High Field Magnet Laboratory (HFML) in Nijmegen, NL, using a custom-built two-axis rotator which allows the sample to be rotated *in-situ* around the polar angle $\theta$ or the azimuthal axis $\phi$ individually. The two-axis rotator was used in one of the $He^4$ flow-cryostats available at the HFML. The temperature is stabilized using the capacitance of a dielectric capacitor, which is known to have negligible field dependence above 4 K (refs 35,36).

The sample was cooled in zero field at a rate of 0.5 K per min to prevent thermal shocks, and the $c$-axis resistivity was measured as the sample was rotated around the polar axis, from $\theta = 0°$ (field normal to crystal planes) to $\theta = 90°$ (field parallel to the the conduction planes) in a fixed field of 30 T. This measurement was repeated at a range of temperatures above and below the long-range magnetic ordering temperature $T_N = 37.5\,K$, from 4.2 to 44 K. In addition, the field dependence was explored by performing polar rotations at fixed fields of 10–30 T in 5 T increments and at a temperature of 4.2 K.

Please see the Supplementary Note 1 for further details.

**Data availability.** All relevant data are available from S.G. and N.E.H.

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

## Acknowledgements

We gratefully acknowledge illuminating discussions with A.P. Mackenzie and N. Shannon, and N. Kikugawa for allowing us to reproduce the $PdCoO_2$ data. We also acknowledge the support of HFML-EMFL, a member of the European Magnetic Field Laboratory (EMFL). This work is part of the research program of the Foundation for Fundamental Research on Matter (FOM), which is part of the Netherlands Organization for Scientific Research (NWO).

## Author contributions

S.G. and N.E.H. initiated the project. S.G., S.L., S.A. and P.R. performed the measurements, H.T. grew the high-quality crystals, S.G., M.I.K. and N.E.H. analyzed the data and wrote the manuscript, with input from all the other authors.

## Additional information

**Competing interests:** The authors declare no competing financial interests.

**Publisher's note**: 

