## [Peer Review File · Nature Communications]

Reviewers' comments:

Reviewer #1 (Remarks to the Author):

The layered metallic oxides PdCrO₂ and PdCoO₂ have been attracting much attention due to their interesting transport properties.

The authors studied interlayer conductivity of PdCrO₂ by measuring angle-dependent magnetoresistance (ADMR) at high fields.

Carefully analyzing the Hanasaki peak, they came to a conclusion that the interlayer incoherence originates from the loss of long-range magnetic order.

Their result will have strong impact on the field of layered materials and will inspire further study for the interlayer coherence.

Their conclusions seem to be robust and reliable with decent data quality and data analysis.

I recommend this article for the publication in Nature Communications, provided that the authors will take into account the following notes and will revise the manuscript.

(i) At which azimuthal angle relative to the crystalline axis did the authors perform the measurement? I could not find any description about this. This information is necessary so that other researcher can reproduce the results. It is known that for PdCoO₂ the azimuthal angle dependence is very huge (PRL 111, 056601). I assume PdCrO₂ would show the similar behavior.

(ii) In the caption of Fig 1, I have a problem for the citation.

Ref 3 and 4 are cited for the sentence "showing the Fermi surface reconstruction at the Neel temperature".

Ref 4 is an ARPES study, in which the Fermi surface reconstruction was not observed.

The dHvA work by J.M. Ok et al. is the first paper discussing the reconstruction, followed by Noh (ARPES) and Hicks (dHvA).

Also, I am not comfortable with the citation style throughout this manuscript.

It starts with a citation to Ref 8, instead of Ref 1.

Reviewer #2 (Remarks to the Author):

This paper describes a series of magnetoresistance measurements on PdCrO₂. PdCrO₂ is a metallic antiferromagnet; the bandstructure is quasi-two-dimensional. The point of this paper is to establish the conditions under which the interlayer transport is coherent, i.e. under which the Fermi surface is three dimensional. Using the observation of a so-called Hanasaki peak or S.Q.U.I.T. in the angle-dependent magnetoresistance, the authors show conclusively that the interlayer coherence is destroyed as the temperature rises through the Neel temperature. Using comparisons with a non-magnetic, isostructural material, PdCoO₂, and a variety of other criteria, the authors conclude that it is the antiferromagnetic fluctuations that drive the destruction of the interlayer coherence.

The paper is carefully written and the result is important, given the number of interesting materials (e.g. cuprate superconductors) where quasi-two-dimensional electronic properties coexist with antiferromagnetic fluctuations. The paper therefore deserves to be published.

There are a few lacunae that the authors should address. The statement "it is still not known what actually triggers interlayer coherence in any layered metal" is untrue, as reference to the many

cited papers by McKenzie and others on organic molecular metals will show. The statement needs to be made more specific to the situation under discussion, where some kind of fluctuations are induced by the collapse of a magnetic ground state.

In view of the fact that there are methods for calculating the field dependence of the Hanasaki peak from the Fermi-surface topology, it should be possible to deduce something about the antiferromagnetic fluctuation rate. As the field increases, theory leads one to expect that antiferromagnetic fluctuations will be suppressed. This may be detectable in the field dependence of the Hanasaki peak. This possibility should be discussed, in view of data from the cuprates where AF fluctuations may affect the amplitude of the de Haas-van Alphen oscillations. In any case, the Neel temperature should be depressed by the field, affecting the temperature of the dimensionality crossover.

Reviewer #3 (Remarks to the Author):

This manuscript describes an angle-dependent magnetoresistance (MR) study on the layered magnetic metal PdCrO₂ and its close analogue, non-magnetic PdCoO₂. These materials are quite topical because of their combination of strong anisotropy, high conductivity and the availability of very clean crystals. PdCrO₂ is of particular interest because it provides one of the best possibilities to study local magnetic moments embedded in a quasi-2d metal. The current manuscript compares MR results on the two materials and concludes that magnetic fluctuations above T_N make interlayer conduction incoherent in PdCrO₂ while nonmagnetic PdCoO₂ retains 3d coherent electron transport. This conclusion, if justified, could be relevant for loss of coherence in other systems (including high T_c cuprates) and as such could be of wide interest.

I believe the measurements have been carefully executed and the results are interesting; if the conclusion is taken at face-value this would be significant enough to justify publication in Nature Communication. However, the authors' interpretation (and the main conclusion of the paper) hinges on the statement that a particular feature of the MR called the Hanasaki peak, is a "definitive and sensitive probe of electron coherence" (L58). While I agree that to observe such a peak requires coherent interlayer transport, the converse is not true i.e. the fact the peak is seen to vanish close to T_N does not imply loss of coherent interlayer transport - I expand on this below. Thus I do not feel that the manuscript is suitable for publication in Nat Comm as it stands. However, if the authors can include arguments for how they can eliminate the less exotic explanations for the loss of the Hanasaki peak, then I will be happy to reconsider my objection. I also include a list of more minor points to be addressed.

1. Hanasaki peak in the Magnetoresistance

The Hanasaki peak is an orbital magnetoresistance effect explained in Refs 20 and 24. It explains the appearance of a sharp resistivity maximum when magnetic field is applied parallel to the conducting planes of a quasi-2d metal. The semi-classical explanation is that the Lorentz force effectively averages out the carrier velocity component v_{\perp} , for a fraction of the electronic carriers, so they cannot contribute to interlayer conduction. This applies both to closed cyclotron orbits (ref 20) and open orbits (ref 24). The sharp reduction of the peak as field-angle moves away from the planes is understood because closed orbits for a tilted field do not generally average v_{\perp} to zero. The effect is very large in PdCoO₂ as shown in the manuscript and already explained in Takatsu et al PRL 111 056601(2013) where the large MR was not described as a Hanasaki peak and the (polar) angle-dependence was not discussed. In that paper it was also argued convincingly that the large size of the effect was due to the crystal quality (so $\omega_c \tau \gg 1$ everywhere) and the sharpness of the corners of the hexagonal Fermi surface giving very few Lorentz-force-free carriers for certain B-directions.

Given that the Hanasaki peak depends on fine details of the Fermi surface and on crystal quality, the lack of a peak above T_N in PdCrO₂ does not constitute strong evidence for loss of interlayer coherence and can plausibly be explained by even small differences in the electronic structure and/or significant differences in carrier lifetime; such less exotic explanations would need to be eliminated before recourse to a loss of interlayer coherence due to magnetic fluctuations. It is also possible that magnetic fluctuations do play a role, but a simpler one than envisaged in the manuscript - namely that they reduce the carrier phase-lifetime via small angle scattering (which would not be captured in the estimate of $\hbar\tau$ from the magnitude of the resistivity in L185); this could wipe out the peak by reducing $\omega_c\tau$. Note that this latter mechanism, though also involving dephasing scattering by magnetic fluctuations is distinct from the conclusion of the paper that magnetic fluctuations destroy coherent propagation perpendicular to the planes.

That a Hanasaki peak is seen below T_N , also does not require an exotic explanation, since PdCrO₂ is effectively a different metal with a different bandstructure and small reconstructed Fermi surfaces for $T < T_N$. This would certainly be expected to give rise to different orbital MR for fields less than the magnetic gaps.

In summary, I feel that as the manuscript stands, the weight of evidence is really not enough to support the main conclusion of the paper. If the less exotic explanations can be eliminated, or at least properly addressed, then this paper could become suitable for publication in Nat Comm.

2. Inset to Fig 2b (and generally throughout) the symmetry ($T > T_N$) of both materials is rhombohedral so there are more c-axis warpings than the simple one considered. This may be an uninteresting complication but could be important for the orbital MR.

3. L57: "the so-called Hanasaki coherence peak". I am not sure that there is precedent for referring to this orbital MR effect as a "coherence peak"; I prefer an objective descriptive term e.g. Hanasaki magnetoresistance peak.

4. L106: "the peak arises due to formation of closed orbits parallel to the conducting plane". While the Hanasaki paper (Ref 20) does claim this, there were different views at the time and Fig 1 of Ref 24 shows that open orbits are as important in forming the maximum in R.

5. L175: "we obtain $t_{\text{perp}} \approx 17 \text{ meV} = 200 \text{ K}$ ". This value could be compared with that from quantum oscillation results and/or band structure calculations. (I think it is encouragingly similar)

6. L108: Figure 2a -> 2b

7. Fig 3a: Can the 4.2K curve be scaled down to see it over the same angle range?

8. L220: "expression in" -> is

Response to comments of Reviewer #1

(review comments in blue/italic)

#1(i) At which azimuthal angle relative to the crystalline axis did the authors perform the measurement? I could not find any description about this. This information is necessary so that other researcher can reproduce the results. It is known that for PdCoO₂ the azimuthal angle dependence is very huge (PRL 111, 056601). I assume PdCrO₂ would show the similar behavior.

The azimuthal angle is such that the applied field rotates from the [001] to the [110] direction. The direction of the azimuthal angle with respect to the crystalline axis had been indicated in Figure 2b of the original manuscript, but was not specified in the main text. We have now included this explicitly in the main text by modifying lines 74–75 to read: “from the field perpendicular ($\theta = 0^\circ$, $H \parallel [001]$) to parallel ($\theta = 90^\circ$, $H \parallel [110]$) to the conduction planes”. The Supplemental Information has also been modified in the same way.

#1(ii) In the caption of Fig 1, I have a problem for the citation. Ref 3 and 4 are cited for the sentence “showing the Fermi surface reconstruction at the Neel temperature”. Ref 4 is an ARPES study, in which the Fermi surface reconstruction was not observed.

The dHvA work by J.M. Ok et al. is the first paper discussing the reconstruction, followed by Noh (ARPES) and Hicks (dHvA).

Also, I am not comfortable with the citation style throughout this manuscript. It starts with a citation to Ref 8, instead of Ref 1.

We thank the reviewer for pointing out the mistake in the citations in the figure caption of Figure 1. The citations have now been modified as proposed by the reviewer. The citations for the sentence “... the Fermi surface reconstruction at the Neel temperature ...” are now the works by Ok *et al.*, Noh *et al.* and Hicks *et al.*

The ordering of citations has now been changed, so that the main text starts with reference [1].

Response to comments of Reviewer #2

(reviewer comments in blue/italic)

#2.1) There are a few lacunae that the authors should address. The statement “it is still not known what actually triggers interlayer coherence in any layered metal” is untrue, as reference to the many cited papers by McKenzie and others on organic molecular metals will show. The statement needs to be made more specific to the situation under discussion, where some kind of fluctuations are induced by the collapse of a magnetic ground state.

As recommended, we have made the statement “it is still not known what actually triggers the loss of interlayer coherence in a number of layered systems ...” in the abstract more specific. It now reads “it is still not known what actually triggers the loss of interlayer coherence in a number of layered systems with strong magnetic fluctuations, such as cuprates and ruthenates.”

#2.2) In view of the fact that there are methods for calculating the field dependence of the Hanasaki peak from the Fermi-surface topology, it should be possible to deduce something about the antiferromagnetic fluctuation rate. As the field increases, theory leads one to expect that antiferromagnetic fluctuations will be suppressed. This may be detectable in the field dependence of the Hanasaki peak. This possibility should be discussed, in view of data from the cuprates where AF fluctuations may affect the amplitude of the de Haas-van Alphen oscillations. In any case, the Neel temperature should be depressed by the field, affecting the temperature of the dimensionality crossover.

The reviewer raised here an interesting point about the field-dependence of the antiferromagnetic fluctuation rate and its possible influence on the field-dependence of the Hanasaki peak. In light of the reviewer’s comment, we have compared the field-dependence of the amplitude of the Hanasaki peak in PdCrO₂ with that measured in non-magnetic PdCoO₂. We find that to within our experimental uncertainty, the field-dependence in both systems is quadratic in field. This would suggest therefore that the AFM fluctuation rate does not vary appreciably within this field range. With regards the cuprates, where clearly AFM fluctuations are present, their influence on the magnetotransport, vis-à-vis the field-dependence of the magnetoresistance, is found to be negligible (see, for example, PRL 75 1395 (1995)) though it is likely to have a significant impact on its temperature dependence.

In the revised manuscript, we have added a short discussion on this point in the Supplemental Information, where we have also included a new two-panelled figure showing that the quadratic field-dependence of the Hanasaki peak in both PdCrO₂ and PdCoO₂.

Regarding the second point, *i.e.* the field-induced suppression of the Neel temperature, we note here that the previous de Haas-van Alphen study up to 33 T of Ok *et al.* (PRL vol. 111, 176405 (13)) showed absolutely no change in the quantum oscillation spectrum up to 29 K, showing that even at these high fields, the magnetic order remains intact. Indeed, given that the exchange energy in PdCrO₂ has been estimated from spin susceptibility measurements to be of order 500 K, it is reasonable to expect that much higher fields would be required to significantly suppress T_N than are employed in our experiments. In the revised manuscript, we have added a sentence (in parentheses) addressing this point together with a reference.

Response to comments of Reviewer #3

(reviewer comments in blue/italic)

*#3.1a The Hanasaki peak is an orbital magnetoresistance effect explained in Refs 20 and 24. It explains the appearance of a sharp resistivity maximum when magnetic field is applied parallel to the conducting planes of a quasi-2d metal. The semiclassical explanation is that the Lorentz force effectively averages out the carrier velocity component v_{\perp} , for a fraction of the electronic carriers, so they cannot contribute to interlayer conduction. This applies both to closed cyclotron orbits (ref 20) and open orbits (ref 24). The sharp reduction of the peak as field-angle moves away from the planes is understood because closed orbits for a tilted field do not generally average v_{\perp} to zero. The effect is very large in PdCoO₂ as shown in the manuscript and already explained in Takatsu *et al* PRL 111 056601(2013) where the large MR was not described as a Hanasaki peak and the (polar) angle-dependence was not discussed. In that paper it was also argued convincingly that the large size of the effect was due to the crystal quality (so $\omega_c \tau \sim 1$ everywhere) and the sharpness of the corners of the hexagonal Fermi surface giving very few Lorentz-force-free carriers for certain B-directions.*

The reviewer refers in particular to the report of Takatsu *et al.* (PRL vol. 111, 056601 (13)) and argues that there, the large MR was not described as a Hanasaki peak. It is important to realize however that the data reported in the Takatsu paper were taken with the magnetic field aligned approximately 3° away from the conducting plane (see Figure 4b and its caption in the Takatsu paper). As shown in Figure 2d of our paper and in Figure 5b of Kikugawa *et al.* (Nature Comm. vol. 7, 10903 (16)), the Hanasaki peak is in fact suppressed completely once the field is rotated 2° away from the plane. Thus, the sharp peaks in the Takatsu data are not associated with the Hanasaki peaks but presumably arise from the band structure effect referred to by the reviewer. Thus the effect described by the reviewer has a different origin, and can be eliminated as an alternative explanation for the disappearance of the Hanasaki coherence peak at T_N . We do not believe any revision of the main manuscript is required to address this point. However, we have added a reference to the Takatsu paper in our discussion of the polar AMRO in the revised Supplemental Information.

#3.1b Given that the Hanasaki peak depends on fine details of the Fermi surface and on crystal quality, the lack of a peak above T_N in PdCrO₂ does not constitute strong evidence for loss of interlayer coherence and can plausibly be explained by even small differences in the electronic structure and/or significant differences in carrier lifetime; such less exotic explanations would need to be eliminated before recourse to a loss of interlayer coherence due to magnetic fluctuations.

Regarding the carrier lifetime, this point was addressed in the original manuscript, where we compared the value of $\omega_c \tau$ in PdCrO₂ at the temperature where the Hanasaki peak vanished with the corresponding value in a quasi-2D organic salt and also with the value at which (polar) peaks disappear in the interlayer magnetoresistance of an overdoped cuprate. It is further instructive to compare the value of the *resistivity* in PdCrO₂ (at T_N) with that in PdCoO₂ at around 90 K. In both cases, this is the temperature where the Hanasaki peak is found to vanish. Given that at T_N , the carrier densities (and effective masses) of the two systems are essentially identical, the ratio of their resistivities should correspond to the ratio of their scattering rates (or inverse carrier lifetimes). According to careful measurements by Hicks *et al.* (PRB vol. 92, 014425 (15)), the resistivity in PdCrO₂ at T_N is approximately twice that of PdCoO₂ at 90 K (see plot below), implying that the scattering rate in PdCrO₂ is about half the value in PdCoO₂ when the Hanasaki peak vanishes. Thus the suppression of the Hanasaki peak cannot be attributed simply to the carrier lifetimes reaching a certain threshold. We have added this additional discussion comparing PdCrO₂ and PdCoO₂ to the main text and thank the reviewer for raising this important point.

#3.1c It is also possible that magnetic fluctuations do play a role, but a simpler one than envisaged in the manuscript - namely that they reduce the carrier phase-lifetime via small angle scattering (which would not be captured in the estimate of \hbar/τ from the magnitude of the resistivity in L185); this could wipe out the peak by reducing $\omega_c \tau$. Note that this latter mechanism, though also involving dephasing scattering by magnetic fluctuations is distinct from the conclusion of the paper that magnetic fluctuations destroy coherent propagation perpendicular to the planes.

Regarding the issue of small-angle scattering, it should be recalled that angle-dependent magnetoresistance (ADMR) is a transport property, and as such, any effects associated with small-angle scattering would not be able to degrade the ADMR features in the same way that they cannot degrade a charge current effectively. This is best illustrated in the paper by Hussey *et al.* (Nature, vol. 425, 813 (03)) where the polar ADMR in the interplane magnetoresistance of an overdoped cuprate could be fitted using precisely the same $\omega_c\tau$ value that is obtained from in-plane Hall effect measurements. This is in contrast, say, to quantum oscillations measured by the de Haas-van Alphen effect where the magnitude of $\omega_c\tau$ is determined by all scattering events, both small-angle and large-angle. In response to this comment, we have added a short section explaining this difference in the Supplemental Information.

#3.1d That a Hanasaki peak is seen below T_N , also does not require an exotic explanation, since PdCrO₂ is effectively a different metal with a different bandstructure and small reconstructed Fermi surfaces for $T < T_N$. This would certainly be expected to give rise to different orbital MR for fields less than the magnetic gaps.

As the reviewer points out, below T_N , PdCrO₂ undergoes Fermi surface reconstruction which gives rise to a different form for the ADMR, specifically the polar ADMR obtained by rotating the field from perpendicular to parallel to the conducting planes. Inspection of the polar ADMR in PdCrO₂ (Figure 2b of our manuscript) and PdCoO₂ (Figure 5a of the Kikugawa paper) shows how sensitive the form of the ADMR is on the band structure, in particular the size of the Fermi cylinders. The Hanasaki peak, on the other hand, is similar in form in both systems. Indeed, to be visible, the Hanasaki peak requires only the existence of a three-dimensional Fermi surface (provided that $\omega_c\tau$ is large enough) and its width is determined uniquely by the ratio between k_F and t_{perp} . It is therefore only weakly dependent on other details of the band structure. In response to the reviewer's comment, we have added a short section emphasising this difference in the Supplemental Information.

In summary, I feel that as the manuscript stands, the weight of evidence is really not enough to support the main conclusion of the paper. If the less exotic explanations can be eliminated, or at least properly addressed, then this paper could become suitable for publication in Nat Comm.

We hope that the reviewer is satisfied that our main conclusions are now more robust and that the loss of the Hanasaki peak at T_N in PdCrO₂ cannot easily be explained by effects not related to the loss of interlayer coherence.

The reviewer also made a number of minor comments that we now turn to.

#3.2 Inset to Fig 2b (and generally throughout) the symmetry ($T > T_N$) of both materials is rhombohedral so there are more c -axis warplings than the simple one considered. This may be an uninteresting complication but could be important for the orbital MR.

The inset in Figure 2b was included simply as a schematic, to illustrate the origin of the Hanasaki peak, and was not intended to be representative of the c -axis warping in PdCrO₂. Nevertheless, it was shown in Hicks *et al.* (PRL, vol. 109, 116401 (12)) that while the c -axis warping in PdCoO₂ was composed of three principal components, the simple sinusoidal warping term was by far (by at least one of magnitude) the dominant term and would therefore be the one primarily responsible for the Hanasaki peak. We believe the same will apply to PdCrO₂. In the revised manuscript, we have emphasised the fact that this inset is just a schematic for illustrative purposes.

#3.3 L57: "the so-called Hanasaki coherence peak". I am not sure that there is precedent for referring to this orbital MR effect as a "coherence peak"; I prefer an objective descriptive term e.g. Hanasaki magnetoresistance peak.

Both McKenzie & Moses (PRL, vol. 81, 4492 (98)) and Singleton *et al.* (PRL vol. 88, 037001 (02), PRL vol. 99, 027004 (07)) refer to the Hanasaki peak as definitive evidence for interlayer coherence. Indeed, in Singleton (PRL vol. 99, 027004 (07)), the term "coherence peak" is used explicitly. Therefore, we believe there is enough precedent to justify the use of the term in our paper. A reference to the Singleton paper has now been added to the main text.

#3.4 L106: "the peak arises due to formation of closed orbits parallel to the conducting plane". While the Hanasaki paper (Ref 20) does claim this, there were different views at the time and Fig 1 of Ref 24 shows that open orbits are as important in forming the maximum in R .

While it is true that the open orbits give a contribution to the interlayer conductivity, Hanasaki *et al.* showed that the open orbits do not lead to the formation of the coherence peak (maximum) at 90°. Specifically, in that paper, Hanasaki *et al.* state that "In the low-field regime, the contribution of the open orbits is larger than that of the small closed ones. However, it does not give the local maximum in the conductivity to cause the clear kink in the lobe of the resistance peak at 90°". Moreover, they state "As the field increases, the relative contribution of the small closed orbits increases." Thus, the statement in our original manuscript is in fact consistent with the conclusions of Ref. [24]. In response to the comment, however, we have added a sentence (in parentheses, L110–113) in the main text reflecting the fact that open orbits can also give a contribution to the interlayer conductivity for in-plane fields.

#3.5. L175: "we obtain $t_{\text{perp}} \approx 17 \text{ meV} = 200 \text{ K}$ ". This value could be compared with that from quantum oscillation results and/or band structure calculations. (I think it is encouragingly similar)

This is a good suggestion. In the revised text, we have added the comparison for t_{perp} from quantum oscillation experiments. As it happens, the latter is found to be of a similar order of magnitude (21 meV). A sentence to this effect has been added to the revised manuscript.

#3.6 L108: Figure 2a -> 2b

We thank the reviewer for pointing out the mistake in the reference to Figure 2. This has now been modified, and refers to Figure 2b.

#3.7 Fig 3a: Can the 4.2K curve be scaled down to see it over the same angle range?

The 4.2 K curve has now been scaled down over the same angle range in Figures 3a and 2c.

#3.8 L220: "expression in" -> is

We thank the reviewer for pointing out the mistake. L220 has now been changed to read "expression is".

Summary of changes to manuscript

#1(i) Lines 74–75 of the main text has modified to specify the azimuthal angle with the phrase "from the field perpendicular ($\theta = 0^\circ$, $H \parallel [001]$) to parallel ($\theta = 90^\circ$, $H \parallel [110]$) to the conduction planes". The Supplemental Information (in section 'Angle-dependent magnetoresistance measurements') has also been similarly modified to contain this information.

#1(ii) The citations in the figure caption of Figure 1 have been modified. The citations for the sentence "... the Fermi surface reconstruction at the Neel temperature ..." are now the works by Ok *et al*, Noh *et al* and Hicks *et al*. The ordering of citations in the main text has also been changed, so that the main text starts with reference [1].

#2.1) The wording of the abstract has been modified to read "it is still not known what actually triggers the loss of interlayer coherence in a number of layered systems with strong magnetic fluctuations, such as cuprates and ruthenates."

#2.2) A new section, entitled 'Field-dependence of the Hanasaki peak' has been added to Supplemental Information addressing the issue of the field-dependence of the AFM fluctuation rate along with a new two-panelled figure showing that the field-dependence of the Hanasaki peak in both PdCrO_2 and PdCoO_2 follows the same quadratic field dependence.

We have also added a new sentence (L118–121 in the revised manuscript) reflecting the fact that the applied field has a negligible effect on T_N at the field strengths employed in this study.

#3.1a A paragraph has been added to the section entitled 'Angle-dependent magnetoresistance measurements' in the Supplemental Information discussing the interlayer magnetoresistance study of Takatsu *et al*. and its relevance, or lack thereof, to the present manuscript.

#3.1b An additional section, comparing the resistivity values of PdCrO_2 and PdCoO_2 at the temperature where their Hanasaki peak is found to vanish has been added to the main text (inserted in L183 of original manuscript). The two paragraphs in this section of the main text have been slightly modified in order to make the comparison clearer.

#3.1c A paragraph has been added to the section entitled 'Estimate of \hbar/τ and $\omega_c\tau$ ' in the Supplemental Information discussing the difference between the damping of de Haas-van Alphen oscillations and angle-dependent magnetoresistance and small- versus large-angle scattering.

#3.1d A short section has also been added to the section entitled 'Angle-dependent magnetoresistance measurements' in the Supplemental Information emphasising the difference between the sensitivity of the polar ADMR and the Hanasaki peak to Fermi surface reconstruction.

#3.2 In the figure caption for the inset in Figure 2b, we have emphasised the fact that the inset is just a schematic for illustrative purposes.

#3.3 A reference to Singleton *et al*. PRL vol. 99, 027004 (07), has been added to the main text.

#3.4 A sentence has been added (in parentheses, L110–113 in the revised manuscript) reflecting the fact that open orbits can also give a contribution to the interlayer conductivity for in-plane fields.

#3.5 A comparison of the estimate for t_{perp} from the width of the Hanasaki peak and from quantum oscillation experiments has been added to the revised manuscript.

#3.6 The reference to Figure 2 on L108 has now been modified and now refers to Figure 2b.

#3.7 In Figures 2c and 3a, the 4.2 K curve has now been scaled down over the same angle range.

#3.8 The typo on line L220 has been corrected.

REVIEWERS' COMMENTS:

Reviewer #1 (Remarks to the Author):

The authors responded properly to my comments. I recommend this article for the publication in Nature Communications.

Reviewer #2 (Remarks to the Author):

I am satisfied with the changes made in response to the referees' comments. The paper should be published.

Reviewer #3 (Remarks to the Author):

I am happy that the authors have considered the points raised and therefore I recommend this for publication.